# LARGE SCALE REPRESENTATION LEARNING FROM TRIPLET COMPARISONS

## ABSTRACT

In this paper, we discuss the fundamental problem of representation learning from a new perspective. It has been observed in many supervised/unsupervised DNNs that the final layer of the network often provides an informative representation for many tasks, even though the network has been trained to perform a particular task. The common ingredient in all previous studies is a low-level feature representation for items, for example, RGB values of images in the image context. In the present work, we assume that no meaningful representation of the items is given. Instead, we are provided with the answers to some triplet comparisons of the following form: Is item A more similar to item B or item C? We provide a fast algorithm based on DNNs that constructs a Euclidean representation for the items, using solely the answers to the above-mentioned triplet comparisons. This problem has been studied in a sub-community of machine learning by the name "Ordinal Embedding". Previous approaches to the problem are painfully slow and cannot scale to larger datasets. We demonstrate that our proposed approach is significantly faster than available methods, and can scale to real-world large datasets. Thereby, we also draw attention to the less explored idea of using neural networks to directly, approximately solve non-convex, NP-hard optimization problems that arise naturally in unsupervised learning problems.

## 1 INTRODUCTION

It has been widely recognized that deep neural networks (DNN) provide a powerful tool for representation learning (Bengio et al., 2013). Representations learned in an unsupervised fashion have been demonstrated to be useful in learning tasks such as classification (Ranzato et al., 2007; 2008; Hinton & Salakhutdinov, 2008; Hinton et al., 2006; Bengio et al., 2007). In the context of supervised learning, representations are typically learned as by-products in neural networks (Radford et al., 2015). For example in image classification, low level representations of inputs (e.g., rgb values) are fed to a network, together with class label information, the network is trained to perform some supervised classification. As a by-product it discovers a condensed data representation in the last hidden layers of the network that turns out to be surprisingly successful for other computer vision tasks such as object detection or semantic segmentation (Girshick et al., 2014; Kümmerer et al., 2014; Long et al., 2015; Ren et al., 2015). Subsequently, more direct mechanisms have been designed to explicitly learn data representations (Mensink et al., 2012; Bell & Bala, 2015; Schroff et al., 2015). Again the raw data is fed to a network, this time together with some information about the similarity of objects, and then the network is trained to generate a meaningful data representation.

Particularly relevant to our work is the field of contrastive representation learning (Wang et al., 2014; Hoffer & Ailon, 2015; Cheng et al., 2016; Ge, 2018; Arora et al., 2019), where similarity information is provided in terms of contrastive triplets of points $(x, x^+, x^-)$: for a given point $x$, the point $x^+$ and $x^-$ are specifically chosen data points that are similar / dissimilar to $x$. Such approaches have been extremely successful on image and text data, and they are particularly elegant if the similarity information can be extracted from the raw data in some unsupervised manner. Examples are the case of word embeddings (Mikolov et al., 2013), where words are considered similar if they occur within the same local neighborhood, or text representations (Logeswaran & Lee, 2018), where two subsequent sentences are considered similar, or in computer vision (Wang & Gupta, 2015) where pairs of image patches in subsequent frames of videos are considered similar.

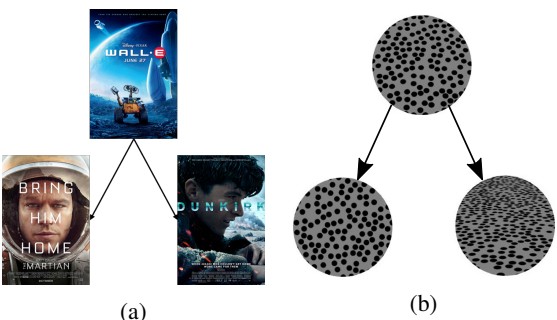

(a)                    (b)

Figure 1: Illustration of general triplet comparisons.

There are two main ingredients that seem unavoidable in all these approaches: First, one needs an explicit low level representation of the data to feed as input to the DNN architectures. Secondly, one needs information about the similarity of different data points that can be used to train the network parameters. This information can be provided by class membership labels in supervised approaches, or it is extracted from some unsupervised meta-data — for instance the temporal information for frames of video data.

In our work, we investigate the problem of learning a data representation in cases where no meaningful input representation or explicit similarity information exists. Instead we assume that we are provided with the answers to a set of general triplet comparisons where, for any arbitrary triplet of items $(x_i, x_j, x_k)$, we know whether $x_i$ is more similar to $x_j$ or $x_k$. The problem of learning a representation based on such triplet comparisons is called ordinal embedding (Agarwal et al., 2007; van der Maaten & Weinberger, 2012; Kleindessner & von Luxburg, 2014; Terada & von Luxburg, 2014).

This setting can be of advantage in many cases. Consider learning a representation for a large set of movies. There does not exist an obvious informative low level representation that can be used in a straight forward manner in a neural net, yet one can ask triplet comparisons from users of a movie database. An example of a triplet comparison on movies in shown in Figure 1a. Other use-cases also arise in more scientific contexts. For example, psychophysical scaling aims to find the functional relation of a physical stimulus and human perception. For example, Aguilar et al. (2017) conducted experiments with triplet comparisons to find out the relation between the angle of a tilted plane and the perceived angle of a human observer. In this study, a number of triplet questions are asked from human observers (see an example of triplet question in Figure 1b). Later, a one-dimensional embedding (representation) is learned for the perceived value of angle. A comprehensive study on the application of ordinal embedding in psychophysics is available at Haghiri et al. (2019).

We design a novel neural network architecture that can learn representations when we are only given the answers to a set of triplet comparisons and no input representation exists. Our approach opens doors into two different worlds:

(1) ***We use the power of DNNs to approximately solve an NP hard optimization problem with discrete input (point identifiers and binary comparisons).*** Its widely believed amongst neural network practitioners that the non-convex landscape of deep and wide neural networks is primarily free of sub-optimal local minima. This theory is supported under simplified assumptions by various theoretical findings (Nguyen & Hein, 2017; Choromanska et al., 2015; Kawaguchi & Bengio, 2019; Kawaguchi et al., 2019). These results provide a basis for the hypothesis that deep neural network models offer a tool that can make non-convex, NP-hard optimization problems "tractable in practice".

In machine learning, this line of thinking is particularly interesting for unsupervised learning problems such as clustering, dimensionality reduction or representation learning, where the typical objective functions are discrete, often NP hard, or non-convex. The standard approach to solve such problems are convex relaxations. Albeit the resulting relaxed problem then can be solved exactly, there often does not exist any provable guarantee that relates the solution of the relaxed problem to the solution of the original problem. We believe that the ability of DNN's to directly (approximately) solve non-convex optimization functions provides an attractive alternative and has not really been

explored to solve such problems in ML: very little emphasis has been placed in approaches that use neural networks not as learning machines but rather as toolboxes to solve optimization problems.

The same point of view might be valuable for general, even discrete optimization problems. In our paper we demonstrate that DNNs can be successfully applied to such a problem. It will be interesting to see how more generic architectures can be designed to solve more generic discrete, non-convex, NP-Hard optimization problems.

(2) ***We provide the first, scalable approach for the ordinal embedding problem based on neural networks.*** The problem of ordinal embedding has been studied extensively in the machine learning literature (Agarwal et al., 2007; van der Maaten & Weinberger, 2012; Kleindessner & von Luxburg, 2014; Terada & von Luxburg, 2014; Jain et al., 2016). Optimizing the ordinal embedding objective by traditional means is notoriously difficult (either algorithms are very slow or lead to unsatisfactory results), and for computational reasons it is close to impossible to embed more than 10000 items. Thanks to the fast parallel computations of DNN training, our approach runs significantly faster than traditional methods (see Subsection 4.4).

## 2 BACKGROUND ON COMPARISON-BASED MACHINE LEARNING AND ORDINAL EMBEDDING

Assume we are given a set of abstract items $X = \{x_1, x_2, \ldots, x_n\}$, but no explicit representation is available for the items. Even though there exists a dissimilarity function $\delta(.,.)$ that describes the dissimilarity of pairs of items, we also do not have direct access to this dissimilarity function. Instead, we only get to see answers to **triplet questions** of the form "Is item $x_i$ more similar to item $x_j$ or item $x_k$? We denote such a triplet question by $t = (x_i, x_j, x_k)$ (or sometimes just by $t = (i, j, k)$), and the answer to this triplet question by

$$R_t = \begin{cases} 1, & \text{if } x_i \text{ is more similar to } x_j \text{than to } x_k \\ -1, & \text{if } x_i \text{ is more similar to } x_k \text{ than to } x_j \end{cases} \qquad (1)$$

Machine learning based on such qualitative comparisons has become popular (Agarwal et al., 2007; van der Maaten & Weinberger, 2012; Amid & Ukkonen, 2015; Ukkonen et al., 2015; Balcan et al., 2016; Haghiri et al., 2017; 2018). One fundamental approach in this area is the **ordinal embedding** procedure. Here we assume that a set of triplet comparisons $T = \{t_1, t_2, \ldots t_m\}$ is given to us. We consider the passive setting where we do not have any influence on the chosen comparisons (as opposed to an active setting where we could actively ask particular questions to an oracle). The binary answers, according to the above definition, are stored. Ordinal embedding aims to find a $d$-dimensional representation $y_1, y_2, \ldots y_n \in \mathbb{R}^d$ such that the Euclidean distances between these points satisfy as many triplet answers as possible. This can be expressed by the following optimization problem:

$$\max_{y_1, \ldots, y_n \in \mathbb{R}^d} \sum_{t=(i,j,k) \in T} R_t \cdot \text{sign}(\|y_i - y_j\|^2 - \|y_i - y_k\|^2). \qquad (2)$$

In a real world setting it is typically impossible to satisfy all the input triplets: the dissimilarity function is not necessarily constrained to be a $d$-dimensional Euclidean metric, and even if so the answers might be noisy.

The above optimization problem is discrete, non-convex and NP-hard (Bower et al., 2018; Pardalos & Vavasis, 1991). There exit various methods that employ relaxations of the this optimization problem: Generalized non-metric multidimensional embedding (GNMDS; Agarwal et al., 2007), stochastic triplet embedding (STE; van der Maaten & Weinberger, 2012) and local ordinal embedding (LOE; Terada & von Luxburg, 2014) are among the methods in this category. However, all of them are computationally very demanding, and none of these methods can scale beyond a few thousand input points at most. The only somewhat scalable approach is the one by Anderton & Aslam (2019), based on explicit geometric constructions and landmarks. However, this approach requires an active access to triplet answers: rather than just receiving a bag of triplet answers, the algorithm needs to repeatedly ask very specific triplet questions to an oracle. Our approach works in the more general passive setting.

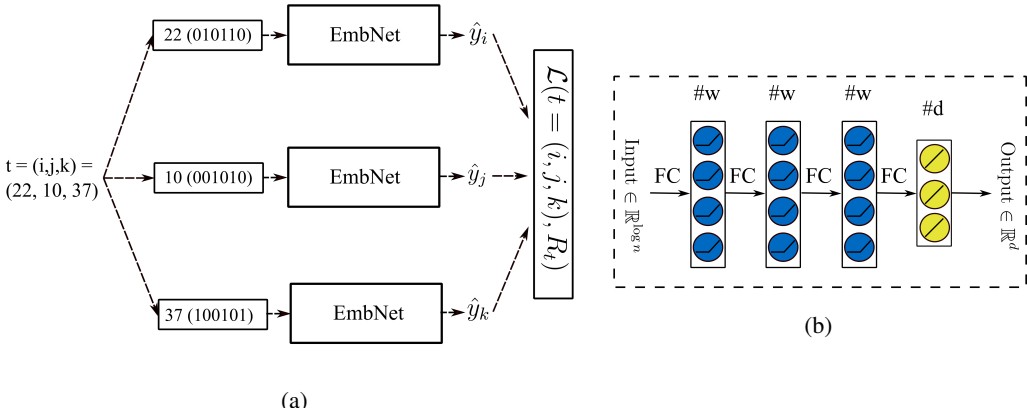

(a)

(b)

Figure 2: (a) The architecture of Ordinal Embedding Neural Network (OENN). As example a triplet $(22, 10, 37)$ and its answer $R_t$ are fed to the architecture. (b) The EmbNet neural network, which is used as a building blocks of ordinal embedding architecture.

The performance of ordinal embedding methods is typically evaluated by the *triplet error*. Given an estimated representation $\{\hat{y}_1, \hat{y}_2, \ldots, \hat{y}_n\}$ of the data items and a set $T'$ of triplet answers, the triplet error simply counts how many triplet answers are satisfied by the given representation:

$$\text{triplet error (TE)} = \frac{1}{|T'|} \sum_{t=(i,j,k) \in T'} \mathbf{1} \left[ R_t \cdot \text{sign}(\|\hat{y}_i - \hat{y}_j\|^2 > \|\hat{y}_i - \hat{y}_k\|^2) = -1 \right], \quad (3)$$

where $\mathbf{1}(\cdot)$ equals to one, if the inside expression is true and it is zero otherwise. Note that $R_t \cdot \text{sign}(\|\hat{y}_i - \hat{y}_j\|^2 > \|\hat{y}_i - \hat{y}_k\|^2) = -1$ if the estimated embedding is not consistent with the triplet $t$. Thus, we count the ratio of non-consistent triplets in this way.

## 3    PROPOSED METHOD: ORDINAL EMBEDDING NEURAL NETWORK (OENN)

Our proposed architecture is inspired by the recent line of work on contrastive learning (Wang et al., 2014; Schroff et al., 2015; Cheng et al., 2016; Hoffer & Ailon, 2015). Figure 2a shows a sketch of our proposed network architecture. The central sub-module of our architecture is what we call the embedding network (***EmbNet***): one such network takes a certain encoding of a single data point $x_i$ as input (typically, an encoding of its index $i$, see below) and outputs a $d$-dimensional representation $\hat{y}_i$ of data point $x_i$. The EmbNet is replicated three times with shared (identical) parameters. The overall OENN network now takes the ***indices*** $(i, j, k)$ corresponding to a triplet $(x_i, x_j, x_k)$ as an input. It routes each of the indices $i, j, k$ to one of the copies of the EmbNet, which then return the $d$-dimensional representations $\hat{y}_i, \hat{y}_j, \hat{y}_k$, respectively (cf. Figure 2a).

The three sub-modules are trained jointly using the triplet hinge loss, as described by the following objective function:

$$\mathcal{L}(t = (i, j, k), R_t) = \frac{1}{|T|} \sum_{t=(i,j,k) \in T} \max \left( -R_t(\|\hat{y}_i - \hat{y}_j\| - \|\hat{y}_i - \hat{y}_k\|) + 1, 0 \right) \quad (4)$$

***Choice of input representation (encoding):*** Since we don't have access to any informative low-level input representations, the choice of input representations presents a challenge to this approach. However, we leverage the expressive power of neural networks (Leshno et al., 1993; Barron, 1993) and their ability to (for instance) fit random labels to random inputs (Zhang et al., 2016) to motivate our choice of input encoding. Since our main goal is to find representations that minimize the training objective, we believe that completely arbitrary input representations are a viable choice.

One such input representation could be the one-hot encoding of the index (where point $i$ is encoded by a string $\hat{x}_i$ of length $n$ such that $\hat{x}_i(l) = 1$ if $l = i$ and 0 otherwise). The advantage of choosing such a representation is that it is memory efficient in the sense that there is no need to additionally

store the representations of the items. However, under this choice of representation the length of the input vectors grows linearly with the number $n$ of input items. As one of the central contributions of our work is to perform ordinal embedding in large scales, we consider a more efficient way: we represent each item by the binary code of its index, leading to a representation length of $\log n$. Such a representation retains the memory efficiency of the one-hot encoding (in the sense as discussed above) but improves the length of the input representation from $n$ to $\log n$. As we will see below, this representation works well in practice. However, note that there is nothing peculiar about this choice of binary code. Indeed, we conducted experiments where each input point was represented by a unique, random binary string of length $\log n$ (see subsection 4.1). The results are pretty similar to the ones using the binary codes. We believe that any representation that dedicates a unique code to the items and covers the $\mathbb{R}^{\lceil \log n \rceil}$ space will do the job.

***Structure of EmbNet:*** Figure 2b shows the schematic of the EmbNet. We propose a simple network with three fully-connected hidden layers and ReLu activation function. The final layer is a linear layer that takes the output of third hidden layer and produces the output embedding. The input size to the network is $\lceil \log n \rceil$ and each hidden layer contains $w$ nodes. The output layer has $d$ nodes to produce embeddings in $\mathbb{R}^d$. The input and output size, $\lceil \log n \rceil$ and $d$, are pre-determined by the task. Thus the only independent parameter of the network is the width $w$ of hidden layers. Our experiments (see subsection 4.1) demonstrate that the hidden layer width ($w$) should grow logarithmically with respect to the number of items $n$ in order to produce good embedding outputs.

**Difference to previous constrastive learning approaches:** As described earlier, our architecture is inspired from Wang et al. (2014); Hoffer & Ailon (2015). However, there are fundamental differences between the works in both the problem that we try to address and our approach to solve this problem. (i) The most stand-out aspect of our problem is that we have no access to representations for the input items $x_1, .., x_n$. Our network takes completely arbitrary representations for the input items. In addition, the way in which triplets are sampled is different as well. (ii) The nature of triplet comparisons is quite different in comparison with the previous studies. In the previous work, triplets are always contrastive triplets of the form $(x, x^+, x^-)$. Obtaining contrastive triplets requires extra information, either either class labels, or more explicit similarity information between objects. In our ordinal framework, we are given neither of this information, and simply gather triplets answers involving arbitrary sets of three points.

# 4 SIMULATIONS

We run an extensive set of simulations, on the one hand to find good rules of thumb to set our hyper-parameters and input encoding, and on the other hand to evaluate the performance of the OENN in terms of triplet error and compare it to alternative ordinal embedding approaches.

## 4.1 CHOICE OF HYPER-PARAMETERS.

Our neural network model consists of a single hyper-parameter, the width of the hidden layers $w$. We ran simulations to inform our choice of $w$ based on the number of input items $n$ and the embedding dimension $d$. Our simulations demonstrated that the width of the hidden layers needs to grow logarithmically with the number of items and linearly with the embedding dimension. The detailed experimental setups as well as the results from these experiments are provided in the supplementary.

## 4.2 CHOICE OF THE LENGTH OF INPUT ENCODING.

The input to the OENN, as described earlier, can be chosen as a triplet of arbitrary encoding of the items. We ran simulations to establish the dependence of the length of such encoding with the number of items. These simulations show that the size of the input encoding needs to grow logarithmically with the number of items. For more details on the parameters of our simulations and the corresponding plots, please refer to our supplementary.

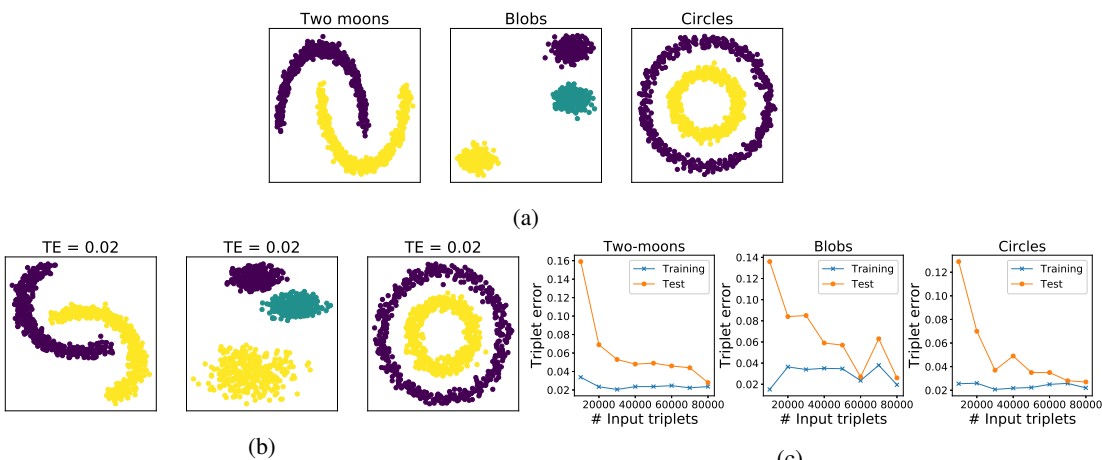

(a)

(b)                                          (c)

Figure 3: (a) Three datasets used for the reconstruction experiment. (b) Embedding output of OENN with $|T| = 80000$ input triplet answers. The training triplet error ($\text{TE}_{train}$) is written on top of each plot (c) Training and test error of the OENN algorithm with respect to varying number $|T$ of input triplet answers.

## 4.3 RECONSTRUCTION AND GENERALIZATION TO UNSEEN TRIPLETS

To demonstrate the capacity of OENNs to reconstruct the true embedding[1], we use three 2-dimensional datasets generated with the `scikit-learn` package in Python. Datasets are shown in Figure 3a: 1- two-moons dataset with two labels, 2- Blobs dataset, which is a mixture of three Gaussians, and 3- concentric circles with 2 labels. In all three cases 1000 items are generated. The label information is added to check the reconstruction capability visually.

The number of input triplets to the embedding algorithm is chosen around $nd \log n \approx 20000$, which is the theoretical bound for the reconstruction of items. More precisely the following range is chosen: $|T| \in \{10000, 20000, \ldots, 80000\}$. We generate the triplet answers according to the following procedure: We first sample three data points without replacement from the given set of $n = 1000$ points, and then evaluate the corresponding triplet question based on the true Euclidean distances between these points. In this experiment, we report the training triplet error ($\text{TE}_{train}$) and test triplet error ($\text{TE}_{test}$). We generate 1000 triplet answers as test set, independent of the training set.

We depict the embedding output of our proposed algorithm with the maximunm number of triplets (80000) in Figure 3b. We reported $\text{TE}_{train}$ on top of each plot. In case of all three datasets, we observe that the output embedding matches the ground truth data closely. In addition, we report the training error and test error of the embedding with various number of input triplets in Figure 3c. The training error is always about $0.03$, showing that the embedding method can satisfy most of the input triplets. However, with few input triplets (10000) the test error is as high as $0.12$. Note that the number of constraints on the ordinal embedding problem increase with the number of triplets and with fewer triplets its easier to find many configurations of data points that satisfy the given set of triplets. The test error, on unseen triplets, however can be higher since the embedding algorithm is less likely to have converged to the true, unique solution up to isometric transformations.

## 4.4 COMPARISON WITH OTHER ORDINAL EMBEDDING METHODS

Here we compare the performance of the proposed OENN method with two competitors, namely local ordinal embedding (LOE) (Terada & von Luxburg, 2014) and t-distributed stochastic triplet embedding (TSTE) (van der Maaten & Weinberger, 2012). The main advantage of proposed OENN method is its ability to scale. In this section we perform a simple simulation with increasing number of items. The items are sampled from a uniform distribution on the unit square in $\mathbb{R}^2$. In a first simu-

---

[1]It has been theoretically shown that an ordinal embedding can be uniquely determined up to an isometric transform — if the items are sampled from a compact subset of the Euclidean space and the number of items grows to infinity (Kleindessner & von Luxburg, 2014)

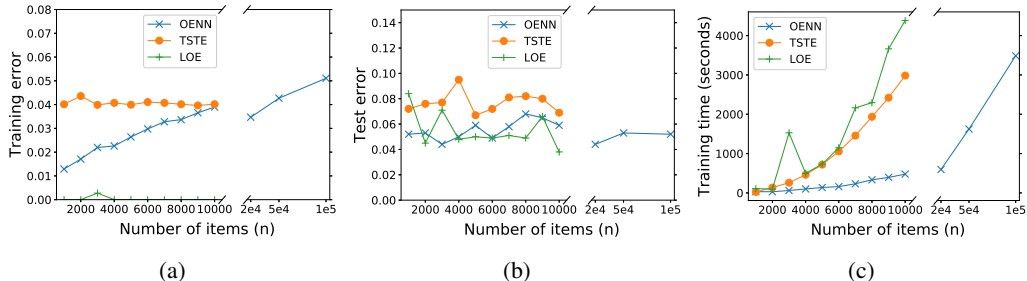

(a)            (b)            (c)

Figure 4: (a) Training triplet error with varying number of items for three different ordinal embedding methods. (b) Test triplet error for varying number $n$ of items. Note that we scale the number of input triplets as $|T| = nd \log n$. (c) Training time (embedding time) of the three methods in seconds. LOE and TSTE did not scale to data sets larger than 10000 points. Note that the right part corresponds to the extended range of items. In addition, the x-axis on the right part is exponential, thus the slope of the curve is not comparable to the left part of the plot.

lation we compare running times for the number of items in the range $n \in \{1000, 2000, \ldots, 10000\}$. Because the running time for LOE and TSTE grows very quickly with $n$, we could not run these two algorithms on larger datasets, while our method is still tractable. Therefore, we run our method alone on an extended range of items: $n \in \{20000, 50000, 100000\}$.

For a fixed set of items we generate $nd \log n$ triplet answers ($d = 2$) based on the Euclidean distances. The answers are fed to the ordinal embedding algorithms as input. For each set of items, we generate an independent set of 1000 triplet answers as test set. The methods are compared based on three evaluation criteria: training triplet error ($\text{TE}_{train}$), test triplet error ($\text{TE}_{test}$), and embedding time in seconds.

For our method, we chose a fixed hidden layer width $w = 400$ for $n <= 10000$ and $w = 800$ for larger $n$. The optimization parameters of LOE and TSTE are set to the default values. The author's implementation of LOE in R, available at CRAN repository[2], is used to perform LOE experiments. We also used the Python implementation of TSTE, available on GitHub[3]. Note that the Python implementation is consistently faster than the original MATLAB implementation by the author. All experiments are performed on the same machine with Intel XEON CPU E5-2620 processor, with 1 GeForce GTX 1080-Ti GPU and 20 GB of memory. Obviously the proposed OENN method takes advantage of fast GPU computations, while the other methods run solely on the CPU.

We depicted the training triplet error and test triplet error of various method in Figure 4a and Figure 4b respectively. LOE can perfectly fit the embedding to the input triplet answers, thus we observe almost zero training error. Our method (OENN) and TSTE both perform reasonably well, showing less than 5% error on the training set. On the test set, all three methods have a similar performance. Finally, we depicted the embedding time of the three methods in seconds on Figure 4c. As the number items grows, the traditional methods (LOE and TSTE) become painfully slow. In case of $n = 1000$, traditional methods require about 1 hour to obtain the embedding, while the OENN requires about 5 minutes. The slope of the curve already shows that, ordinal embedding in large scales in essentially impossible with the traditional methods. In contrast, the OENN method can embed 100000 items in about 1 hour, as shown on the right part of Figure 4c. Note that the scale of the x-axis on the right part is exponential, thus the growth is computation time is not comparable to the left side.

## 5 EXPERIMENTS

In this section, we perform ordinal embedding on a large scale dataset of triplet answers. To obtain such a dataset, we ran a study on Amazon's mechanical Turk (MTurk) platform. The study aims to collect answers to triplet comparisons of natural images. As data set we used a subsample of the Imagenet (Deng et al., 2009) dataset. We chose the three high-level concepts "animal", "food", and

---

[2]https://cran.r-project.org/web/packages/loe/index.html
[3]https://github.com/gcr/tste-theano

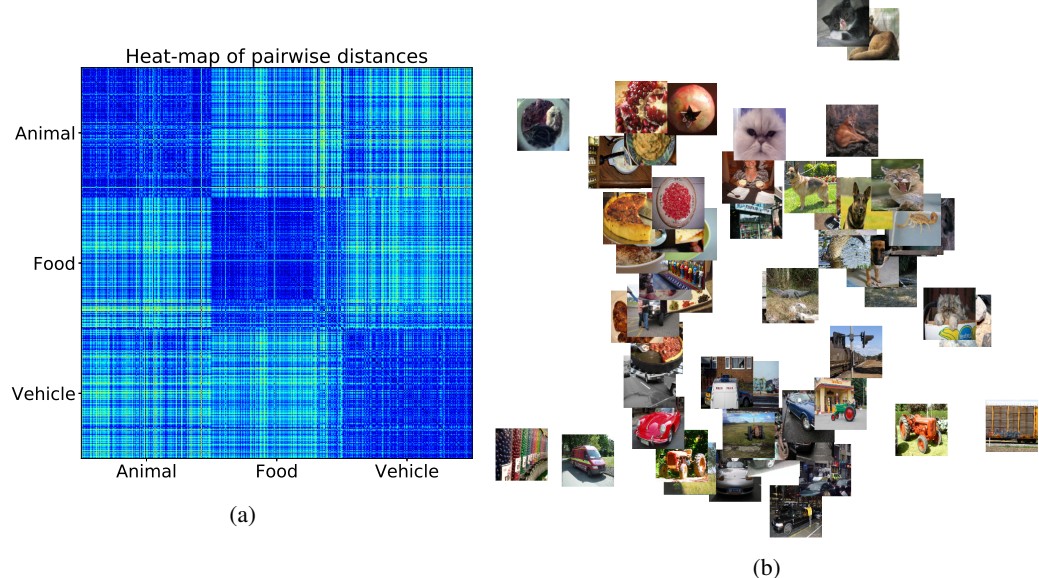

Figure 5: (a) The heat-map of pairwise distances between the embedding outputs. The cencept name for each chunk of 2500 items is written on both axes. (b) An example of 60 images depicted on their actual embedded location in two dimensions. Four images from each category are chosen at random.

"vehicles", and in each of these concepts we picked 5 sub-categories ("synsets"). The details of these 15 categories and of our MTurk implementation are presented in the supplementary material.

We gathered 120000 triplet answers from the MTurk workers, which is somewhat less than the $nd \log n \approx 190000$ triplets that would be required for high-quality recovery according to the theoretical lower bound for an embedding in $d = 2$ dimensions (Jain et al., 2016).

Figure 5a depicts the pairwise distances of embedding outputs with embedding dimension $d = 2$. The three clusters are clearly evident in the distance matrix. We also depict a the embedding output for a subsample of 60 images in Figure 5b. We randomly chose 4 images from each category to produce a sample of 60 images. Looking at the embedding output, we clearly see that three concepts of animals, vehicles and food are located at upper right, bottom, and upper left part of the picture respectively. There are a few exceptions as well. For instance on the very left bottom we see the shelf of a confectionery store.

# 6 DISCUSSION

In this paper, we propose a novel, scalable representation learning algorithm. A unique distinguishing property of our method is that it does not need any feature representation for the items — we successfully apply DNNs with a random input encoding to achieve meaningful representations. As input we only require binary, relative distance comparisons.

Our proposed DNN architecture exploits the ability of neural networks to approximately solve non-convex, NP-hard problems to obtain near optimal solutions to ordinal embedding problems. We believe that this provides a new perspective to DNNs, which opens the doors to apply DNNs on other challenging optimization problems, particularly in the context of unsupervised learning.

In its application domain, our proposed method shows an outstanding improvement in terms of the computation time of ordinal embedding problems. Over the last decade, the ordinal embedding problem has been analyzed exhaustively, and a number of methods have been proposed. However, ordinal embedding for more than thousands of items has been beyond the reach of algorithms. We proposed the first method that can work with passively collected triplet answers and solves the large scale ordinal embedding problem.

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

# A  Appendix

## A.1  Dependence between the layer width $w$ and the number $n$ of items

As the number $n$ of items grows, we expect that a larger network is required to address the complexity of the task. We can either increase the representational power of the network by adding more layers or by increasing the width of layers. In our experience a network with three hidden layers is always sufficient to solve the ordinal embedding problems — if the layer width is chosen properly. Therefore, we focus on a network with three hidden layers and investigate the required number of nodes in each layer (layer width $w$).

In our first simulation, we fix the space $\mathbb{R}^2$, and choose $n$ points from its unit square according to the uniform distribution. We consider $n \in \{2^7, 2^8, \dots, 2^{14}\}$. Given a sample of $n$ points, we generate $nd\log n$ triplet answers (where $d = 2$ is the dimension) according to the following procedure: We first sample three data points without replacement from the given set of $n$ points, and then evaluate the corresponding triplet question based on the true Euclidean distances between these points. The number of triplets is chosen based on the theoretical lower bound on the required number of triplet answers for the recovery of Euclidean representation (Jain et al., 2016), ignoring constants that might have been swallowed in the big-O-notation.

The embedding network is constructed with 3 hidden layers, each having $w$ fully-connected neurons with ReLu activation functions. The width $w$ of the layers (= number of neurons in each layer) is chosen from $w \in \{70, 90, \dots, 170\}$. For a fixed network and a fixed set of triplet answers, the ordinal embedding is executed 10 times in order to examine the average behaviour of the model.

The implementations use the "PyTorch" open-source library. The initial weights of the hidden layers and the final embedding layer are chosen according to a uniform distribution (default initialization of PyTorch). We used Adam(Kingma & Ba, 2014) to optimize the weights of the network. The learning-rate and batch-size parameters of the Adam optimizer are chosen as $5 * 10^{-3}$ and 5000, respectively.

Figure 6a shows the training triplet error ($\text{TE}_{train}$) of the ordinal embedding with varying number of items and width of the hidden layers. The error is reported with a heat-map plot, where warmer colors denote higher triplet error. The result shows that logarithmic growth of the layer width respect to $n$ is enough to obtain desirable performance.

## A.2  Dependence between layer width $w$ and embedding dimension $d$

Besides the number of items, the embedding dimension is another factor that influences the complexity of ordinal embedding. We expect that the required layer width needs to grow with the embedding dimension. We perform a simulation with $n = 1000$ items in a $d$-dimensional Euclidean space with varying dimension $d$. Similar to the previous subsection, we sample 1000 items from the unit cube in $\mathbb{R}^d$. Then, we generate triplet answers to $nd\log n$ triplet questions based on the Euclidean distances of items.

As we hypothesize the that relation between required layer width and the dimension is linear, we choose the dimension in $d \in \{5, 10, \dots, 40\}$. The width of the hidden layer is chosen in the linear range $w \in \{100, 125, \dots, 225\}$. The number of hidden layers, the optimization algorithm and the hyperparameters of optimization are the same as the previous subsection.

Figure 6b shows the training triplet error ($\text{TE}_{train}$) of the ordinal embedding with varying dimension and width of the hidden layers. In order to achieve a desirable triplet error (below 10%), the width of hidden layers need to grow with the dimension. There is again a clear line of transition between low and high error regions. As both axes are chosen linear in this experiment, we can conclude that a linear increase in layer width with respect to the dimension is needed.

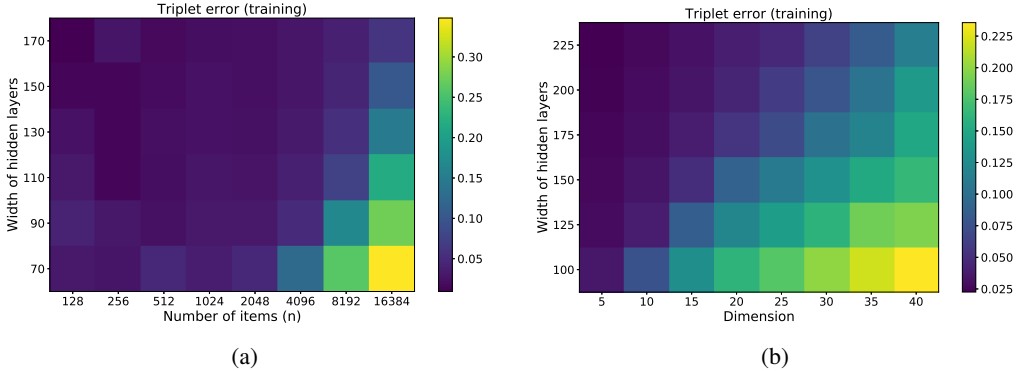

(a)                                                                  (b)

Figure 6: (a) Triplet error with varying number of items and hidden layer size. The x and y axes correspond to the number of items (n) and the hidden layer size (w) respectively. Note that x-axis grows exponentially. (b) Triplet error with varying dimension and hidden layer size. The x and y axes correspond to the number of dimensions (d) and the hidden layer size (w) respectively.

## A.3   DEPENDENCE BETWEEN THE SIZE OF THE INPUT ENCODING $d'$ AND THE NUMBER OF POINTS $n$

Our simulations demonstrate that the size of the input encoding needs to grow logarithmically with the number of points. On the account of space constraints, experimental details and plots from these simulations are available in the supplementary file of this article.

## A.4   DEPENDENCE ON SIZE OF THE INPUT REPRESENTATION

Our Ordinal Embedding Neural Network takes random encoding of a triplets of items as input and learns a transformation from the input encodings to vectors in Euclidean space of a dimension specified by the task. We ran simulations with binary encoding of the indices as well as with input representations of a fixed length ($d'$) by choosing each component of the vector uniformly at random from an arbitrary interval (we choose the interval $[1, 100]$ in our experiments). Our simulations show that the size of the input encoding needs to grow logarithmically with increasing number of items.

To conduct the simulations, we sampled $n$ (ranging in $\{2^7, 2^8, ..., 2^{14}\}$ ) data points uniformly from a unit square in $R^2$. For a fixed set of $n$ items, we generate $nd \log n$ triplet answers (where $d = 2$ is the dimension) according to the procedure as described in the main paper (see section 4.1). Our simulations demonstrated that choosing the width of the hidden layers at the order of $\log n$ would suffice to achieve low reconstruction error. So, we construct our EmNet by choosing the width of the hidden layer sufficiently large ($w = 100$) for $n = 2^{14}$, which is the largest number of items used in our simulation.

Figure7 shows the training triplet error ($\text{TE}_{train}$) of the ordinal embedding with varying number of items and size of the input encoding. The result shows that logarithmic growth of the size of the input encoding with respect to $n$ suffices to obtain desirable performance.

## A.5   RECONSTRUCTION, DETAILED RESULTS

Here, we report the detailed results of the reconstruction experiment, presented in Subsection . Figure 8 shows the true datasets and the estimated embedding with different number of input triplets. The left column shows the true datasets. The number of input triplet is chosen in $r \cdot nd \log n$, where $r \in \{0.5, 1, 2, 4\}$. This number is written on top of each plot, increases from left to right. In case of all three datasets, we observe that the output embedding gets more similar to the ground truth dataset as we increase the number of input triplet answers. However, the loss and the training triplet error are both almost steady. If the algorithm is provided with less triplet answers, it needs to satisfy less constraints, thus it has a simpler task. In other words, there exist more than one solution to the problem. However, in all cases the algorithm can satisfy almost the whole set of input triplet answers ($\text{TE}_{train} \approx 0.03$).

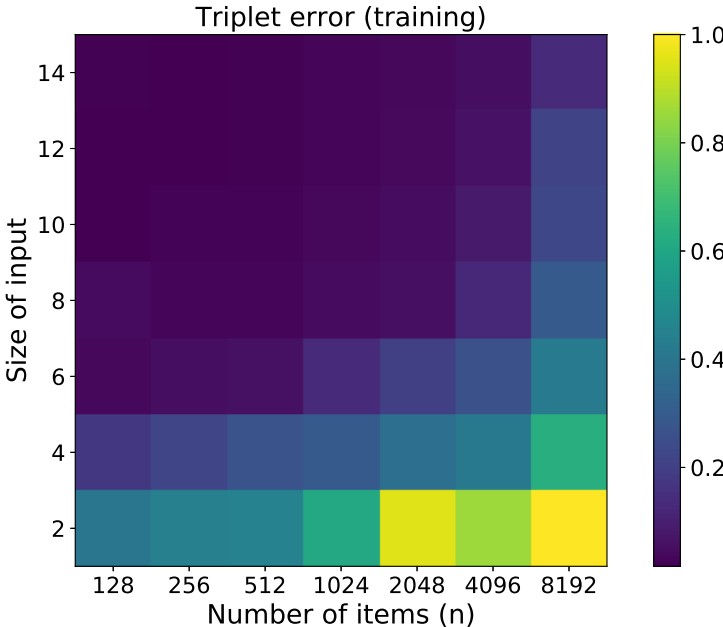

Figure 7: The horizontal axis of the heatmap represents the number of points. Note that it increases exponentially. The vertical axis represents the size of the chosen input encoding. This axis increases linearly. The color of the heat map represents the triplet error obtained in the training procedure.

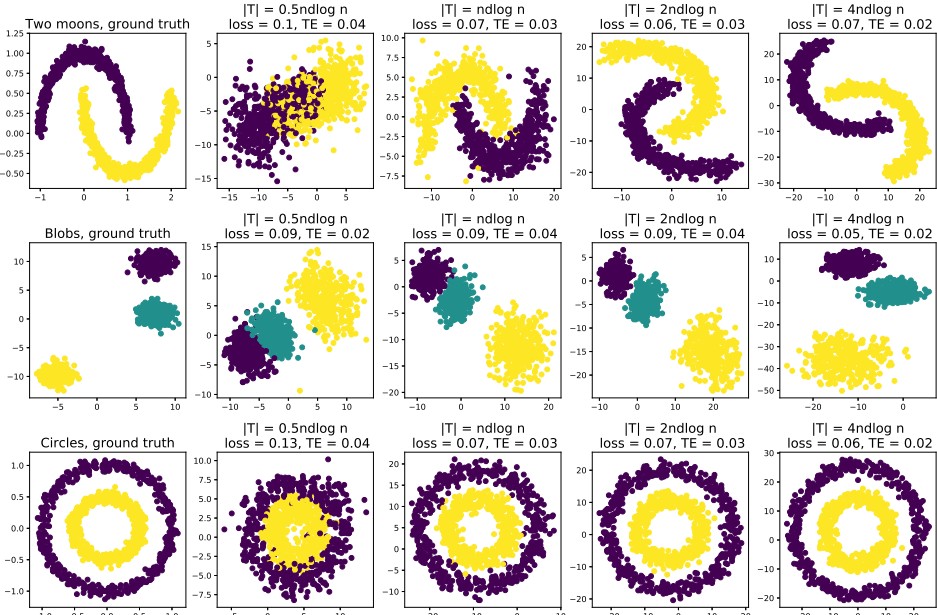

Figure 8: The reconstruction of three toy datasets with various number of input triplet answers. Each column corresponds to one dataset. In each row, the first plot (from left) depicts the original datasets. The four next plots show the embedding output of OENN with various triplet inputs. The number of triplet inputs ($|T|$), training loss, and the triplet error (TE) have appeared on top of each plot. Colors of the dots represent hypothetical labels for the items.

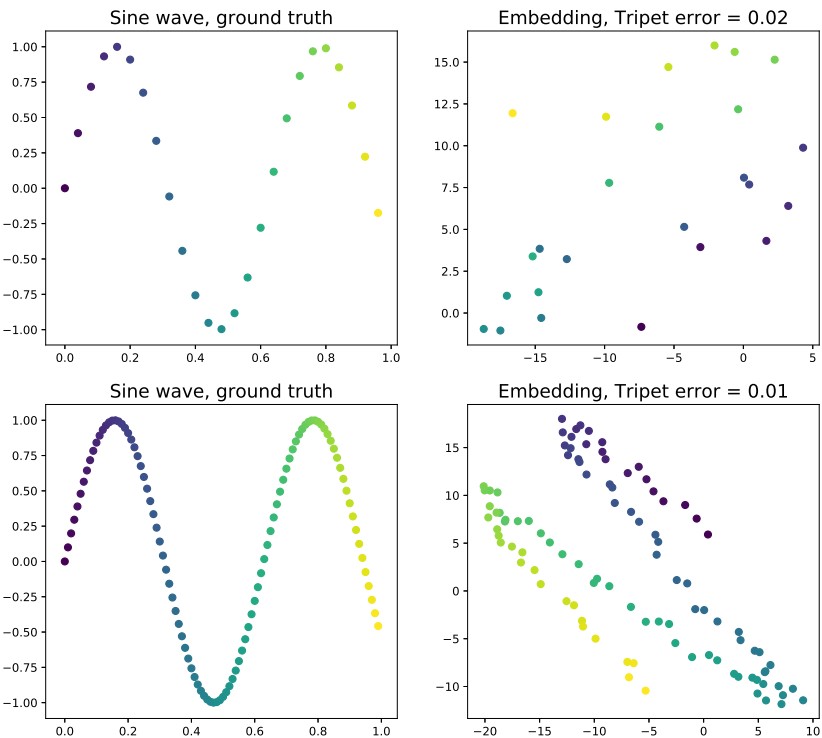

Figure 9: An example of a pathological dataset for ordinal embedding

## A.6 EXTRA DATASETS, PATHOLOGICAL EXAMPLES

In our three examples for the reconstruction experiment the datasets exhibit some nice properties: large number points, clustered data distribution. The natural question would be: "Does the proposed method still work in absence of the nice properties? Do we still get a good reconstruction of the initial dataset?"

Here, we present two examples in which the above properties do not hold. We chose a sine wave as the data distribution. The two datasets have 25 and 100 points respectively (see Figure 9, left column). We perform the ordinal embedding in two dimensions with $nd\log(n)$ triplet answers, which are produced based on Euclidean distances in two dimensions. Figure 9 (right column) depicts the embedding result. The training triplet error for both experiments is still less than $2\%$, which shows that the solution is a valid solution.

Regarding the quality of reconstruction, the embedding with more data points resembles a sine wave better than fewer data point. This is due to the fact that with less number of data points there is more room for the embedding output to wiggle and the output still satisfies all triplet answers. Here, our ordinal embedding method is able to find a valid solution — a solution which satisfies the input triplet answers — irrespective of dataset properties. This solution may not be unique as the ordinal embedding problem does not necessarily have a unique solution. Hence, one cannot expect a perfect reconstruction in such cases.

## A.7 DETAILS OF THE MTURK EXPERIMENT

We conducted the crowd-sourcing study with a web-based implementation of the triplet comparisons on our servers. The workers are redirected to our page, where they have to answer 2000 triplet questions. The chosen concepts are very diverse, thus many resulting triplet questions might be not meaningful. For instance, a sports car cannot really be considered more similar to a dog than to a pizza or vice versa. The workers are advised to give an answer when they can see a similarity, otherwise an option was provided to declare that the comparison cannot be made.

The triplet answers collected from workers can expected to be noisy. Beside the natural sources of human error, some workers may provide random or very unreliable answers. To filter out such workers, we designed a sanity check system. In every set of 2000 triple questions in one session, there exist about 200 triplet questions with very obvious answers. These are the cases where the pivot item $x_i$ is in the same category with one the choices $x_j$, while the other choice $x_k$ has a different category. If a worker has less than 70% accurate answers on these triplets, we rejected his/her work, and consequently did not add the response to the final pool of triplet answers.

The three concepts of { Animal , Food, Vehicle }, each contain 5 categories. The labels of the categories are as follows:

{ Dog, Cougar, Alligator, Scorpion, Cat, Pizza, Pomegranate, Ice cream, Mashed potato, Confectionery, Goods wagon, Tractor, Police van, Limousine, Sport car }

The larger and detailed pairwise distance matrix of two dimensional embedding is shown in Figure 10

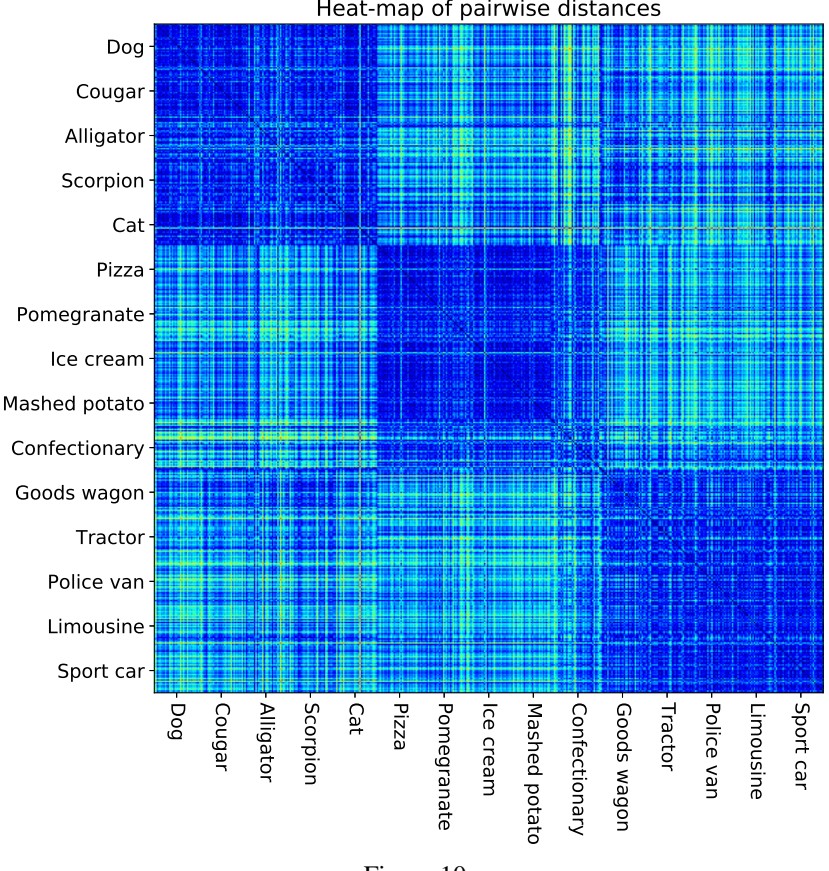

Figure 10

## A.8 THE CHOICE OF THE LOSS FUNCTION

In this section, we justify the choice of the loss function used in the paper. The problem of ordinal embedding - finding an embedding $X = \{x_1, x_2, .., x_n\} \in \mathbb{R}^d$ that satisfies a set of given triplets, $\mathcal{T}$ - can be phrased as a quadratic feasibility problem (Bower et al., 2018) as shown in equation 5.

$$\text{find } X \text{ subject to } X^T P_{i,j,k} X > 0 \ \forall (i,j,k) \in \mathcal{T}. \tag{5}$$

Each $P_{i,j,k}$ corresponds to a triplet constraint that satisfies,

$$||x_i - x_j||^2 > ||x_i - x_k||^2 \iff X^T P_{i,j,k} X > 0$$

Every feasible solution to 5 is a valid solution to the problem of ordinal embedding.

An equivalent way to solve 5, i.e., find a feasible solution of 5 is by finding the global optima of the constrained optimization problem (Bower et al., 2018) given by 6.

$$\min_{X \in \mathbb{R}^{nd}} \sum_{(i,j,k) \in \mathcal{T}} \max \left\{ 0, 1 - X^T P_{i,j,k} X \right\} \tag{6}$$

Meaning, every feasible solution to (5) can be scaled to attain global optima of (6) and every global optima of (6) is a feasible solution of (5) (Bower et al., 2018). Moreover, in optimization (1), any positive scaling of a feasible point $X$ is a solution as well. Whereas in optimization (2) this effect is eliminated.

To summarize, the hinge loss satisfies some nice properties in the sense that using the hinge loss to solve the ordinal embedding problem is not a relaxation but rather an equivalent one.