# OpenReview forum: "LARGE SCALE REPRESENTATION LEARNING FROM TRIPLET COMPARISONS"
_ICLR.cc/2020/Conference — Reject_

### Official Review · AnonReviewer1 · 2019-10-24
**Official Blind Review #1**

**Rating:** 3

**Review:**

The paper proposes to use the triplet loss as a convex relaxation of the ordinal embedding problem. The loss is solved using feed-forward neural network with the input to the network being the ids of the items encoded in binary codes. The benefit of using a deep network is to exploit its optimization capability and the parallelism on GPUs. The experiments presented in the paper include a set of simulation experiments and a real-world task.

I am giving a score of 3. This work is an interesting application of deep learning, but it gives little insight as to why deep networks are able to solve the problem and how to solve ordinal embedding itself.

To elaborate, the problem is known to be NP-hard in the worst case, while the data sets used in the paper seem to have certain nice properties. It would be interesting to see how deep networks do for the hard cases. It would also be interesting to see if additional assumptions, such as the existence of clusters or separation between clusters, make ordinal embedding simpler and thus tractable. Another approach is to assume the solution to have low surrogate loss (4), and any convex solver with sufficiently large number of points is able to find such a solution. Then the question becomes how deep networks solve the particular convex optimization problem. Thinking along these directions would bring more insight and impact to both the ordinal embedding problem and optimization in deep networks.

one quick question:

equations (3) and (4)
--> isn't this the same as using the hinge loss to bound the zero-one loss?


**Experience Assessment:**

I have read many papers in this area.

**Review Assessment: Checking Correctness Of Derivations And Theory:**

I carefully checked the derivations and theory.

**Review Assessment: Checking Correctness Of Experiments:**

I carefully checked the experiments.

**Review Assessment: Thoroughness In Paper Reading:**

I read the paper thoroughly.

---

> ### Author Response · Authors · 2019-11-13
> **Author reponse**
>
> We would like to thank the reviewer for their feedback. We address each comment below individually with appropriate headings.
>
> - Summary
>
> We would like to point out that the reviewer in the summary incorrectly described that our approach uses the "triplet loss as a convex relaxation of the ordinal embedding problem". Using the triplet loss as a proxy does not make the problem convex.
>
> - The relation between data distribution and hardness of ordinal embedding
>
> Ordinal embedding is NP-hard independent of the data distribution. The paper “Landscape of non-convex quadratic feasibility” (Bower et al. 2018) can shed more light on this. The equation (1) in this paper rephrases the ordinal embedding problem as a homogeneous quadratic feasibility problem. The constraint matrices of the problem (P_i in the paper), which correspond to the triplet inequalities, are all indefinite which makes the whole optimization NP-hard.
>
> Moreover, many of our experiments in this paper feature the uniform distribution, which does not satisfy any nice structural assumptions.
>
> - Using a convex solver
>
> As we pointed out earlier, using the triplet loss does not make the optimization problem convex and hence using a convex solver would not be possible here.
>
> - “Equations (3) and (4):  isn't this the same as using the hinge loss to bound the zero-one loss?”
>
> Yes, that is true.

---

> > ### Comment · AnonReviewer1 · 2019-11-14
> > **Clarification**
> >
> > Thanks for the response.
> >
> > Yes, I agree the entire problem is not convex when we optimize the coordinates of all the points. This was my mistake, but you can introduce auxiliary variables to leverage the convexity, and do an iterative, EM type of update. My point is that the loss function has certain nice properties, and I am expecting the paper to have more development on those.
> >
> > I do not agree that the problem is NP-hard regardless of the data distribution. You can construct pathological distributions that makes the problem easy to solve if you know the distribution a priori. Regardless, my point is that maybe there are certain properties/constraints that make the problem easier to solve, and maybe the experimental settings happen to satisfy these properties/constraints. That's why I mentioned hard examples. Approximation also plays a role here, because we probably do not need to solve the problem exactly. (As least that is the sentiment I get from the experiments.)
> >
> > In terms of insights as to how neural networks solve them, other reviewers seem to have similar issues. I have read the responses, and I agree that this message of the paper is to point out that neural networks are able to solve this particular NP-hard problem. However, I find it weak to have this single message as a paper.
> >
> > Here are some concrete points to improve the paper. I would raise the score if the paper includes them.
> >
> > * Why do the authors think the networks are able to solve the problem? Can you form hypotheses based on those intuitions?
> > * How does the use of constrastive loss play a role in the hypotheses? Is it necessary or is it sufficient? How much do we lose from the relaxation? (This is where the properties of the loss function come in.)
> > * Can you do experiments to confirm or disprove those hypotheses?

---

> > > ### Author Response · Authors · 2019-11-15
> > > **Author response (3/3)**
> > >
> > > 3) Do the distributions satisfy some nice properties and this is why the problem of ordinal embedding is somehow easier which enables neural networks to solve the problem?
> > > It is natural to ask if the data distributions satisfy nice properties which allow the neural network to solve an easier optimization problem. Our short answer to this question is that the proposed method solves the ordinal embedding problem even for datasets that do not possess such nice properties. To provide evidence to support this claim, we refer to experiments conducted on 3 different datasets, which could be considered pathological, to demonstrate that our approach can achieve a low training error even in these cases. 1) Experiments in sections 4.4, A.2, A.3, A.4 are all performed on the uniform distributions which do not possess any pattern. Moreover, we chose 2 extra datasets, to demonstrate that our approach can minimize the objective just as well even in these cases. We refer the reviewer to Figure 9 in Appendix A.6 for more details.
> > > We would like to add some further clarification differentiating the two tasks: 1) Reconstruction of the original dataset and 2) Solving ordinal embedding.
> > > The two problems are not equivalent. The ordinal embedding solution tends to a unique solution --- up to an isometric transform--- when the number of points grows large. For datasets with smaller sample sizes, the solution is not unique. Intuitively one can observe that wiggling points in their position do not violate the triplet answers.
> > > The focus of our work is to solve the ordinal embedding problem. The problem is solely to find a feasible set of points (embedding). However, observe that ordinal embedding is typically an under-determined problem, i.e., the solution is not unique. A simple example to see this would be to consider three points in some Euclidean space and generate all possible triplets from these points. One can verify that several possible configurations of these points satisfy all the triplets.

---

> > > ### Author Response · Authors · 2019-11-15
> > > **Author response (2/3)**
> > >
> > > 2) Why are neural networks able to approximately solve the problem? Can you develop some hypotheses based on these intuitions and some experiments to prove/disprove the hypothesis?
> > > A popular hypothesis among neural network theorists and practitioners is that training a neural network typically leads to a non-convex optimization problem where the quality of local optima is improved toward the global optima with increasing width and depth. There is a lot of theoretical and empirical literature supporting this hypothesis.
> > > For instance, 1* showed that in deep linear networks with the square loss function, the objective function is non-convex in the network parameters and yet every local minima is a global minima. 2* showed that in fully connected neural networks with RELU activation functions, the quality of all differentiable local minima improves with increasing width and depth. In practice, this implies that first-order, gradient-based - approaches such as SGD can be used to efficiently train the networks.
> > > Moreover, 3* also showed that for any fixed input dimension $d$, RELU networks of width $d+1$ and arbitrary depth can approximate any real-valued function on $d$ input variables.
> > > Our primary conjecture is that this would hold true for our architecture as well with respect to our chosen loss function. We provide experimental evidence to support our hypothesis (already in the paper) demonstrating that with increasing width of the hidden layers, the objective value (triplet loss) achieved by the network decreases.
> > > 1* Kawaguchi, Kenji. "Deep learning without poor local minima." Advances in neural information processing systems. 2016.
> > > 2*Kawaguchi, Kenji, Jiaoyang Huang, and Leslie Pack Kaelbling. "Effect of depth and width on local minima in deep learning." Neural computation 31.7 (2019): 1462-1498.
> > > 3*Zhang, Chiyuan, et al. "Understanding deep learning requires rethinking generalization." (2016).

---

> > > ### Author Response · Authors · 2019-11-15
> > > **Author response (1/3)**
> > >
> > > We thank the reviewer for their constructive feedback! We address all the reviewer’s comments below and made necessary revisions to the paper.
> > > 1) What are the nice properties of using the specific loss function and do we lose something by relaxation?
> > > Short answer: Using the hinge loss leads not to a relaxation but an equivalent optimization problem to ordinal embedding. We describe this in detail below. This is established in the ordinal embedding literature and for completeness sake, we also added the below explanation in subsection A.8 in the appendix.
> > > The problem of ordinal embedding - finding an embedding $X = \left \{ x_1, x_2, .., x_n \right \} \in \mathbb{R}^d$  that satisfies a set of given triplets, $\mathcal{T}$ - can be phrased as a quadratic feasibility problem (1) as shown below.
> > > \begin{equation}
> > > 	\textrm{find } X \textrm{ subject to } X^T P_{i,j,k} X > 0 \textrm{ } \forall (i,j,k) \in \mathcal{T}.
> > > \end{equation}
> > > Each $P_{i,j,k}$ corresponds to a triplet constraint that satisfies,
> > > $$\vert \vert x_i - x_j \vert \vert^2  > \vert \vert x_i - x_k \vert \vert^2 \iff X^T P_{i,j,k} X > 0 $$
> > > Every feasible solution to problem above is a valid solution to the problem of ordinal embedding. Note that here we rephrased the same problem as defined in Equation (2) of the main paper.
> > > An equivalent way to solve the above problem, i.e., find a feasible solution that satisfies the constraints is by finding the global optima of the constrained optimization problem (1) given by the optimization problem as shown below.
> > > \begin{equation}
> > > 	\min \limits_{X \in \mathbb{R}^{nd}}  \sum \limits_{(i,j,k) \in \mathcal{T}} \max \left \{ 0, 1 - X^T P_{i,j,k} X \right \}
> > > \end{equation}
> > > Meaning, every feasible solution to the first problem can be scaled to attain global optima of the second one and every global optima of the second problem is a feasible solution of the first (1). Moreover, in the first problem, any positive scaling of a feasible point $X$ is a solution as well. Whereas in the second one, this effect is eliminated.
> > > To summarize, the hinge loss does satisfy some nice properties in the sense that using the hinge loss to solve the ordinal embedding problem is not a relaxation but rather an equivalent one.
> > >
> > > (1) Bower, Amanda, Lalit Jain, and Laura Balzano. "The Landscape of Non-Convex Quadratic Feasibility." 2018 IEEE International Conference on Acoustics, Speech and Signal Processing (ICASSP). IEEE, 2018.

---

### Official Review · AnonReviewer2 · 2019-10-27
**Official Blind Review #2**

**Rating:** 1

**Review:**

The paper presents a way to learn a vectorial representation for items which are only described by triplet similiarity expressions.

The paper not only claims 'large scale representation learning' but also utilizing the described idea to use neural networks to "directly, approximately solve non-convex NP-hard optimization problems that arise naturally in unsupervised learning problems." Both claims are not really shown in the paper: (i) The experiments are not large scale and (ii)  it becomes not clear how any substantiate insight with respect to NP-hard problems can be gained here apart from the fact that it tackles a ML problem, which many seem to be computationally hard problems.

As such the paper is not convincing. On a more detailed level it is not clear why the log n representation for items is choosen -- why not just map to embeddings directly? The more interesting question of how to generalize to unseen items (how would that be possible given that items have no representation at all) is not discussed at all and seems not to be realizable, which makes the starting point of such methods (items have no representation) questionable.

The paper also misses relevant citations of similar questions from the field of (probabilistic) matrix factorization and relational learning.

**Experience Assessment:**

I have read many papers in this area.

**Review Assessment: Checking Correctness Of Derivations And Theory:**

I assessed the sensibility of the derivations and theory.

**Review Assessment: Checking Correctness Of Experiments:**

I assessed the sensibility of the experiments.

**Review Assessment: Thoroughness In Paper Reading:**

I read the paper at least twice and used my best judgement in assessing the paper.

---

> ### Author Response · Authors · 2019-11-13
> **Author response**
>
> Thanks for your feedback. We discuss each comment in the following:
>
> - The experiments are not large scale
>
> We respectfully disagree with the reviewer's main comment that the experiments are not large scale. One needs to see the background of existing work: Existing ordinal embedding methods are known to be notoriously slow and embedding more than 10,000 points is not practical - as reflected in our experiments (see Figure 4). Our new approach manages to get one order of magnitude higher (100000 many points and about 4 million triplets), without diverting to heuristics such subsampling or adding extra information such as invoking active oracles (as needed in landmark approaches). Sure, this is not the scale of 80 million tiny images; but one wouldn’t ask an author of an improved SAT-solving algorithm, say, to scale to 80 million instances.
>
> Representation learning, the topic of this conference, has many facets. Learning representations from “big data” (as in 80 million images with RGB representations) is one side, but learning representations when little data is available (no explicit representation, just binary-valued triplet comparisons) is the other side. Both are valuable in different circumstances.
>
> - No substantiate insight with respect to NP-hard problems
>
> We would like to clarify that our claim was merely that we use neural networks to address ONE instance of an NP-hard optimization problem. We want to bring attention to the generic idea of using neural networks as optimization toolboxes to directly solve non-convex optimization objectives instead of merely for learning problems.
> To elaborate, consider optimization problems that arise in unsupervised learning - for instance, ordinal embedding objectives, clustering objectives or dimensionality reduction objectives. These optimization problems are typically not solved directly since there are non-convex, discrete, NP-hard. Instead, we resort to convex relaxations and many convex relaxations do not come with any guarantees. Consider, however, if we could use a non-convex optimization toolbox to directly tackle the original optimization problem - which is currently NOT the standard practice in ML. Then the value of the true objective already informs us of how close we are to the optimal solution of the optimization problem. So powerful non-convex solvers might be of a significant advantage over convex relaxations. Our paper simply shows ONE example for this.
>
> - It is not clear why the log n representation for items is chosen -- why not just map to embeddings directly?
>
> It would not be possible to set the input dimension the same as the embedding dimension.
>  Our experiments demonstrate that we need input representations of size at least Omega (log n) to sufficiently reduce the triplet error. The size of the embedding dimension can be too low to achieve this. One could argue that instead of using a small network like ours, a heavily over-parameterized neural network could potentially accomplish the same with smaller input representation. However, the computational complexity of the method is significantly affected by this and this is in conflict with the main goal of the paper: scaling ordinal embedding.
>
> - Methods, where items have no representation, are questionable
>
> Items having no representation is a caveat of the data available rather than that of the method. The representationless framework of triplets is relevant to many applications (e.g. crowdsourcing), and the whole field of comparison-based learning works in this framework.
>
> - How to generalize to unseen items
>
> First, it is not standard practice to discuss the generalization to unseen instances in unsupervised machine learning problems, for example in the literature on clustering. But of course, if generalization exists, it is of advantage.
> We believe that in our case, generalization is realizable. One possible approach would be to reserve some extra bits in the binary representation of inputs, and then utilize it to represent new items. The network can be trained with extra batches of triplets which involves the new items.
>
> - The paper also misses relevant citations of similar questions from the field of (probabilistic) matrix factorization and relational learning.
>
> We don’t really see a link to matrix factorization or relational learning. If the reviewer has some idea of such connections, we would be happy to learn of this.

---

### Official Review · AnonReviewer4 · 2019-11-04
**Official Blind Review #4**

**Rating:** 3

**Review:**

Summary:

Many prior works have found that the features output by the final layer of neural networks can often be used as informative representations for many tasks despite being trained for one in particular. These feature representations, however, are learned transformations of low-level input representations, e.g. RGB values of an image. In this paper, they aim to learn useful feature representations without meaningful low-level input representations, e.g. just an instance ID. Instead, meaningful representations are learned through gathered triplet comparisons of these IDs, e.g. is instance A more similar to instance B or instance C? Similar existing techniques fall in the realm of learning ordinal embeddings, but this technique demonstrates speed-ups that allow it to scale to large real world datasets.

The two primary contributions of the paper are given as:
- a showcase of the power of neural networks as a tool to approximately solve NP-hard optimization problems with discrete inputs
- a scalable approach for the ordinal embedding problem

After experimentation on synthetic data, they compare the effectiveness of their proposed method Ordinal Embedding Neural Network (OENN) against the baseline techniques of Local Ordinal Embedding (LOE) and t-distributed Stochastic Triplet Embedding (TSTE). The test error given by the systems is comparable, but there are clear speed benefits to the proposed method OENN as the other techniques could not be run for a dataset size of 20k, 50k, or 100k.

Then, they gathered real-world data using MTurk applied to a subset of ImageNet and applied OENN to learning embeddings of different image instances using only the MTurk triplet information rather than the input RGB input features.

Decision: Weak Reject

1. Interesting technique to take advantage of neural networks to efficiently learn ordinal embeddings from a set of relationships without a low-level feature representation, but I believe the experiments could be improved. One of the main advantages of this approach is efficiency, which allows it to be used on large real-world datasets. The MTurk experiment gives a qualitative picture, but it could be improved with comparisons to pairwise distances learned through alternative means using the RGB image itself (given that images would permit such a comparison). By this I mean, that you may be able to use relationships learned using conventional triplet methods which use input RGB features as ground truth, and test your learned relationships against those. However, since quantitative exploration of large real-world datasets may be challenging and expensive to collect, the synthetic experiments could have been more detailed. The message of synthetic experiments would be stronger if more of them were available and if the comparison between LOE, TSTE, and OENN was made on more of them.

2. I think that the claim that the use of neural networks with discrete inputs can approximately solve NP-hard optimization problems is an exciting one, which likely necessitates more experiments (or theoretical results), but as it stands I don't think it is a fundamentally different conclusion from the fact that this method provides a great scalable solution for the ordinal embedding problem. This claim can be made secondarily or as motivation for continued exploration along this direction, but I think listing them as two distinct contributions is necessary.

Additional feedback:

Since quantitative real-world results are challenging to obtain, improved presentation of the qualitative results would be helpful as well. You may be able to show more plots which help display the quality of the embedding space varying with the number of triplets used. For example, an additional plot after Figure 5 (b) which shows a few scatter plots of points (color coded by class) for training with different numbers of collected triplets. Also, since it should be fairly easy to distinguish between cars and animals or cars and food, it may be more interesting to focus on the heat-maps from along the block diagonal of Figure 5 (a) and talk about what relationships may have been uncovered within the animal or food subsets.

Very minor details:

In Figure 5, a legend indicating the relationship between color intensity and distance would be helpful.

In Figure 6 there seem to be unnecessary discrepancies between the y-axis and colorbar of subplots (a) and (b), and keeping those more consistent would improve readability.

**Experience Assessment:**

I have read many papers in this area.

**Review Assessment: Checking Correctness Of Derivations And Theory:**

N/A

**Review Assessment: Checking Correctness Of Experiments:**

I carefully checked the experiments.

**Review Assessment: Thoroughness In Paper Reading:**

I read the paper at least twice and used my best judgement in assessing the paper.

---

> ### Author Response · Authors · 2019-11-13
> **Author response**
>
> We thank the reviewer for the feedback. In the following, we address the comments individually:
>
> -  Comparison with conventional triplet methods using images and their corresponding RGB images
>
> We did not consider comparisons with conventional triplet approaches: the message of our paper was not to demonstrate the utility of ordinal embedding approaches over conventional (representation-based) triplet approaches.  It was rather to show that when input representations are NOT available, we provide a scalable approach to solve the ordinal embedding problem. We agree with the reviewer that such a comparison would be possible but the experiments, we believe, would not reflect the message of the paper.
>
> -  More synthetic experiments comparing the various ordinal embedding approaches
>
> There is a large literature that compares existing ordinal embedding approaches, and in order to not overload the figures, we had decided to just compare against the most popular traditional algorithms. But we can definitely add more comparisons in the revision of the paper.
>
> -  The “claim that the use of neural networks with discrete inputs can approximately solve NP-hard optimization problems”
>
> We would like to clarify that our claim was merely that we use neural networks to address ONE instance of an NP-hard optimization problem. We want to bring attention to the generic idea of using neural networks as optimization toolboxes to directly solve non-convex optimization objectives instead of merely for learning problems.
> To elaborate, consider optimization problems that arise in unsupervised learning - for instance, ordinal embedding objectives, clustering objectives or dimensionality reduction objectives. These optimization problems are typically not solved directly since there are non-convex, discrete, NP-hard. Instead, we resort to convex relaxations and many convex relaxations do not come with any guarantees. Consider, however, if we could use a non-convex optimization toolbox to directly tackle the original optimization problem - which is currently NOT the standard practice in ML. Then the value of the true objective already informs us of how close we are to the optimal solution of the optimization problem. So powerful non-convex solvers might be of a significant advantage over convex relaxations. Our paper simply shows ONE example for this.
>
> - Additional feedback: 1) scatter plots for the MTurk experiment with an increasing number of triplets 2) detailed analysis of heat-map distance matrix
>
> Both suggestions will be added to the revision to enhance the analysis of the experiment. We will add the scatter plots of the training set (a subsample of the set), color-coded by the category, similar to scatter plots in Fig 3. Moreover, it is certainly possible to consider the pairwise distances heat-map of Figure 5. We plotted a detailed version of this plot in Figure 9, with full category labels. There are indeed meaningful patterns in block diagonals. For instance, we had the "confectionery store" category in the food concept, which is conceptually a bit far from food. Thus, we observe a clear rectangle with warmer colors. This is also the case for the "goods wagon" in the Vehicle concept.

---

### Official Review · AnonReviewer3 · 2019-11-04
**Official Blind Review #3**

**Rating:** 6

**Review:**

The paper presents a Neural Network based method for learning ordinal embeddings only from triplet comparisons.
A nice, easy to read paper, with an original idea.

Still, there are some issues the authors should address:

- for the experiment with Imagenet images, it is not very clear how many pictures are used. Is this number 2500?
- the authors state that they use "the power of DNNs" while they are experimenting with a neural network with only 4 layers. While there is no clear line between shallow and deep neural networks, I would argue that a 4 layer NN is rather shallow.
- the authors fix the number of layers of the used network based on "our experience". For the sake of completeness, more experiments in this area would be nice.
- for Figure 6, there is not a clear conclusion. While, it supports that " that logarithmic growth of the layer width respect to n is enough to obtain desirable performance."  I don't see a clear conclusion of how to pick the width of hidden layers, maybe a better representation could be used.
- I don't see a discussion about the downsides of the method (for example, the large number of triplet comparison examples needed for training; and possible methods to overcome this problem).
- in section 4.4 when comparing the proposed approach with another methods why not use more complex datasets (like those used in section 4.3)
- in section 4.3, there is no guarantee that the intersection between the training set and test set is empty.
- in section 4.3 how is the reconstruction built (Figure 3b)?

A few typos found:
- In figure 3 (c) "number |T of input" should be  "number |T| of input"
- In figure 5 (a) "cencept" should be "concept"
- In figure 8 "Each column corresponds to ..." should be "Each row corresponds to ...".
- In the last paragraph of A1 "growth of the layer width respect" should be "growth of the layer width with respect"
- In the second paragraph of A2 "hypothesize the that relation" should be "hypothesize that the relation".
- In section 4.3 last paragraph, first sentence: "with the maximunm number" should be "with the maximum number"


**Experience Assessment:**

I have published one or two papers in this area.

**Review Assessment: Checking Correctness Of Derivations And Theory:**

I assessed the sensibility of the derivations and theory.

**Review Assessment: Checking Correctness Of Experiments:**

I assessed the sensibility of the experiments.

**Review Assessment: Thoroughness In Paper Reading:**

I read the paper at least twice and used my best judgement in assessing the paper.

---

> ### Author Response · Authors · 2019-11-13
> **Author response**
>
> We thank the reviewer for the insightful comments. We address the questions in the following:
>
> - How many images did you have in the experiment?
>
> We had 7500 images in total. We had 3 concept classes, and 2500 images for each concept. We will mention the total number in the main text.
>
> - The proposed network is not deep, but shallow
>
> We agree that a clear distinction line between shallow and deep networks does not exist. So we will make a note on that issue.
>
> - More experiments on the number of layers
>
> We had experimented with fewer layers. We realized that in this case the width of the network should be increased to compensate for the representation power of the network. As we already had an extensive set of experiments, we decided not to report that. As the proposed architecture already performs well to solve the ordinal embedding problem, we found it unnecessary to try deeper networks.
>
> - "I don't see a clear conclusion of how to pick the width of hidden layers, maybe a better representation could be used."
>
> There exist three parameters in this experiment, which makes it hard to come up with the most conclusive representation. We also generated line plots (multiple curves in one plot) and 3D mesh plots to show the dependency. In the end, we found the heat-map more informative. In the revision, we will add the other plots to support the claim.
>
>
> - "I don't see a discussion about the downsides of the method"
>
> One of the drawbacks is that our method needs GPUs, while the more traditional algorithms run on CPUs. This can be of disadvantage if non-machine learning experts want to use our method. However, this is the case for most recent ML methods based on neural nets.
>
> The number of required triplets is theoretically lower bounded by nd log n, and this is also being confirmed by our experiments (our algorithm, as well as our competitors, break down when they get fewer triplets). Therefore, in a setting with passive triplet answers, and without extra information, it is impossible to overcome this problem.
>
> - "in section 4.4 when comparing the proposed approach with another method why not use more complex datasets (like those used in section 4.3)"
>
> Independent of the dataset complexity, provided with enough triplet answers, all methods can yield less than 5% triplet error. However, the computation time is significantly lower for our proposed method. Due to the iterative nature of all algorithms, the computation time does not depend on the data distribution, but on the number of input points. Thus, a simple uniform dataset could serve to show our intention in this section.
>
> - "in section 4.3, there is no guarantee that the intersection between the training set and the test set is empty."
>
> Yes, in theory that is true, but in practice this is negligible: the total number of possible triplets is about 10^9. So the likelihood that two sets of size 1000 intersect is close to 0.
>
> - "in section 4.3 how is the reconstruction built (Figure 3b)?"
>
> Figure 3b is the exact output of the ordinal embedding in two dimensions. The colors are the initial labels of the input items. There are two or three labels assigned to demonstrate the quality of reconstruction. Note that the ordinal embedding output is unique only up to isometric transforms. In other words, every valid output is still valid with rotation, scaling and translation.

---

### Comment · Area_Chair1 · 2019-11-14
**Reviewers, any comments on the author responses?**

Dear Reviewers, thanks for your thoughtful input on this submission!  The authors have now responded to your comments.  Please be sure to go through their replies and revisions.  If you have additional feedback or questions, it would be great to get them this week while the authors still have the opportunity to respond/revise further.  Thanks!

---

### Decision · Program_Chairs · 2019-12-19

**Decision:**

Reject

**Comment:**

The authors demonstrate how neural networks can be used to learn vectorial representations of a set of items given only triplet comparisons among those items.  The reviewers had some concerns regarding the scale of the experiments and strength of the conclusions:  empirically, it seemed like there should be more truly large-scale experiments considering that this is a selling point; there should have been more analysis and/or discussion of why/how the neural networks help; and the claim that deep networks are approximately solving an NP-hard problem seemed unimportant as they are routinely used for this purpose in ML problems.  With a combination of improved experiments and revised discussion/analysis, I believe a revised version of this paper could make a good submission to a future conference.